# *SCDet*: A Robust Approach for the Detection of Skin Lesions

**DOI:** 10.3390/diagnostics13111824

**Published:** 2023-05-24

**Authors:** Shahbaz Sikandar, Rabbia Mahum, Adham E. Ragab, Sule Yildirim Yayilgan, Sarang Shaikh

**Affiliations:** 1Department of Computer Science, University of Engineering and Technology Taxila, Taxila 47050, Pakistan; shahbaz.sikandar@students.uettaxila.edu.pk (S.S.); rabbia.mahum@uettaxila.edu.pk (R.M.); 2Industrial Engineering Department, College of Engineering, King Saud University, Riyadh 11421, Saudi Arabia; 3Department of Information Security and Communication Technology (IIK), Norwegian University of Science and Technology (NTNU), 2815 Gjøvik, Norway; sule.yildirim@ntnu.no

**Keywords:** convolution neural network, benign, malignant, skin cancer, batch normalization, max pooling, skin lesion, softmax, dermoscopic images

## Abstract

Red, blue, white, pink, or black spots with irregular borders and small lesions on the skin are known as skin cancer that is categorized into two types: benign and malignant. Skin cancer can lead to death in advanced stages, however, early detection can increase the chances of survival of skin cancer patients. There exist several approaches developed by researchers to identify skin cancer at an early stage, however, they may fail to detect the tiniest tumours. Therefore, we propose a robust method for the diagnosis of skin cancer, namely *SCDet*, based on a convolutional neural network (CNN) having 32 layers for the detection of skin lesions. The images, having a size of 227 × 227, are fed to the image input layer, and then pair of convolution layers is utilized to withdraw the hidden patterns of the skin lesions for training. After that, batch normalization and ReLU layers are used. The performance of our proposed *SCDet* is computed using the evaluation matrices: precision 99.2%; recall 100%; sensitivity 100%; specificity 99.20%; and accuracy 99.6%. Moreover, the proposed technique is compared with the pre-trained models, i.e., VGG16, AlexNet, and SqueezeNet and it is observed that *SCDet* provides higher accuracy than these pre-trained models and identifies the tiniest skin tumours with maximum precision. Furthermore, our proposed model is faster than the pre-trained model as the depth of its architecture is not too high as compared to pre-trained models such as ResNet50. Additionally, our proposed model consumes fewer resources during training; therefore, it is better in terms of computational cost than the pre-trained models for the detection of skin lesions.

## 1. Introduction

Biomedical images are used more frequently for the timely recognition, prevention, and classification of deadly illnesses [1]. Cells and skin tissues are assessed for the diagnosis of any illness. Usually, the organs are scanned, for example, X-rays, ECG, and MRIs [2]. Imaging modalities such as X-rays, CT scans, and MRIs are commonly used by oncologists for cancer detection. The structure of the body parts and physiological functions are evaluated by employing these modalities of biomedical imaging [3]. Furthermore, the progression of several types of tumours: brain, lungs, and skin are assessed and detected using CT scans [4]. Whereas adipose nerves and breast tumours can be diagnosed easily using mammography [5,6]. This technique is painful, therefore advanced imaging ways have been introduced for breast cancer diagnosis [7]. Cardiac and gum disease diagnoses are achieved by CT [8,9]. These techniques of biomedical imaging are utilized to detect the above-mentioned diseases, yet we are anxious about skin cancers and how they are detected employing these techniques of biomedical imaging. In Ref. [10], the authors utilized X-rays and CT scans for lung analysis to identify the coronavirus. Similarly, in [11], fundus images are utilized for eye disease detection, namely Glaucoma. 

Brown or red stains over the skin are known as skin lesions [12]. Primary skin lesions exist by birth. Secondary skin lesions are formed by the irregular progress of cells. Skin lesions that are formed by irregular growth of cells are recognized as malignant tumours or cancer. Skin tumours have two types: non-melanoma and melanoma. Melanoma skin cancer may cause the death of a patient in a later stage; however, it is twenty times lesser than other forms of cancer [13]. The annual count for skin cancer patients has enlarged over the past few decades due to a lazy lifestyle and unhygienic food.

Skin cancer manifestation is continuously growing globally [14]. In India, approximately 5000 patients having skin cancer are admitted in a single year [15]. Skin cancer detection requires additional effort, time, and care if performed by eye [16]. Moreover, the grading of tumours is a time-taking process that requires high expertise. The results may be imprecise and necessitate effort and time. The challenges in identifying tumours include asymmetrical shapes and several types exhibiting similar appearances and magnitudes. The erroneous diagnosis may cause severe complications for the patient’s health and reduces survival chance. To address the challenges mentioned above, the room for developing computerized skin cancer detectors has been proposed [13]. Although the biopsy process of histopathology is used for skin cancer detection, it is a painful process and needs to be replaced [17]. Various works have been proposed by researchers for skin melanoma detection, however, they fail to diagnose tiny skin tumours. Therefore, in this study, we propose a robust method for skin cancer detection that can identify the tiniest tumours on the skin with maximum accuracy. 

The important features of this paper are below:To propose an automated robust method to detect skin cancer from skin images with high accuracy;The technique is based on a novel architecture of a convolutional neural network (CNN). We have utilized 32 layers including image input as the first layer, and 30 layers in between including an output layer. This model makes possible an effective detection of benign and malignant skin lesions through these layers;The utilization of a dropout layer has regularization properties that help the *SCDet* to minimize overfitting;We performed a comparative analysis with existing deep learning models, i.e., SqueezeNet, AlexNet, and VGG16. The results show that *SCDet* outperformed the existing techniques;We assessed the performance using the Dermis Dataset containing 1000 samples of skin lesions, distributed into validation, training, and testing sets. The model provides 99.6% accuracy for skin lesion detection and we noticed that the results are remarkable for tiny tumours as well;We analysed that the proposed model utilized less computational cost than existing methods;We also compared the performance of *SCDet* with the existing techniques of machine learning (ML), segmentation, and deep learning (DL) models. We observed that our proposed model provides high performance in terms of accuracy and precision.

The remaining paper is arranged as follows: Section 2 is about the related work that describes the methods of skin lesions that have been developed previously; Section 3 describes the methodology of the proposed CNN and how the model is implemented for the detection of skin lesions; Section 4 demonstrates the experimentation and results of our proposed CNN along with comparison with other state-of-the-art models; and finally, Section 5 is about the conclusion and future work.

## 2. Related Work

Benign lesions can be differentiated from malignant cancer using DL techniques that successfully use visual samples. Asymmetrical appearance mostly represents melanoma or malignant skin lesion. Benign skin lesions have regular edges while lesions are irregular and blurred in the case of melanoma. The shade of grey algorithm is utilized for preprocessing in [18]. Furthermore, for the segmentation of skin lesions, Mask R-CNN is utilized, which is a DL approach. Morphological operations have been performed for the noise removal method in the skin images. Recognition of a lesion appearing on the skin using dermoscopy images, comprising three steps, has been executed in [19]: (1) image preprocessing to enhance the performance of the technique by splitting them into two classes; (2) augmentation of images is used on data to shield the method from overfitting; (3) Densnet-121 is used to mine the features and U-net is proposed for skin cancer identification. However, the results described in the paper do not represent correctly the accuracy of the proposed model.

H. A. Hasan developed a hybrid identification method and converted the images for the recognition of skin lesions. Non-cancerous images were assigned a value of 0 and cancerous ones as 1 into the Numpy array. For the training of CNN using K-fold validation, the dataset was split into two sets to test and train. An 85.303% accuracy was achieved by the Xception net [20], whereas other methods have also been tested, such as mobile net v2, Resnet 50, and VGG19. Mobilenet v2 provided a minimum accuracy of 54.54%. Less accuracy was achieved due to the use of images that were of low quality present in the dataset. Identification of the skin cancer was performed by employing the ML approach Support Vector Machine (SVM) in [21] to detect melanoma. The grey level co-occurrence matrix (GLCM) method was used as a feature descriptor/extractor and extracted features were then fed to SVM for skin cancer detection. The accuracy attained by the model was 95%, however, it can be enhanced by applying some preprocessing methods of image enhancement on the dataset to minimize the noise present in the images and to improve the procedure of training.

M. R. Ibraheem in [22], developed a contrast-limited method of adaptive histogram equalization to improve the lesions of images. Bilinear interpolation was used in Contrast to limited adaptive histogram equalization (CLAHE) along with a threshold equalization algorithm. Moreover, a pixel-based technique was utilized to segment the lesions and extract features. The labels of classes were 0 to 2, background objects were referred to with 0, benign was referred to with 1, and melanoma was denoted by 2. Gradient Boosted Tree (GBT) provided 97.5% accuracy. Rahajeng, M. Nuh in [23] utilized various techniques such as a median filter, cropping, and threshold as preprocessing processes. Additionally, Sobel filters and active contours were used to segment the skin lesions depending on the shape, texture features, and colour achieved using the GLCM algorithm. SVM was used to recognize the category of skin cancer attaining an accuracy of 85%.

In G. S. et al., 400 × 400 pixel resized images are passed to an input layer. To extract features, a 32-filter convolutional layer was employed. Further, reparameterization was utilized using a BN layer to improve the performance by internal covariant shift [24]. A ReLU layer was employed after max-pooling and FC layers. The authors achieved 89.3% accuracy having the loss factor as 0.2633; however, accuracy could be increased by modifying training weights and customizing the network’s layer structure. The authors in [25] utilized CNN to categorize skin images into cancerous and non-cancerous. The authors used the International Skin Collaboration 2016 dataset for training which comprised images with the dimensions 1024 × 767 pixels. These images consisted of three types: melanoma was categorized as cancerous, whereas seborrheic keratosis and nevus were categorized as benign. The generalized Gaussian distribution technique was utilized to segment images with a CNN to classify the images. The accuracy attained for the proposed model was 98.32%, however, the technique was not validated on other datasets.

Y. Filali developed a classification technique by decomposing the images of skin lesions into texture and object components. To attain the region of interest, segmentation was employed on the objects after the texture and segmented area were combined. Feature extraction was performed using Convolution layers. A pooling layer just followed by the convolution layer was employed to minimize the spatial size and then FC layers were applied. The softmax activation function followed by a final classification layer was used as an activation function to classify melanoma as cancerous and nevus as a non-cancerous class. For this method, 93.50% accuracy was attained, which could be improved [26]. N. Rezaoana proposed a CNN model for nine classes: MEL, VASC, BCC, AKIEC, NV, DF, BKL, seborrheic keratosis, and squamous carcinoma. Data augmentation operations such as flip, rotation, and shear were used on the images. Conv. layers were utilized to extract features, a max-pooling layer was used to minimize the dimensions, and a softmax was utilized after the pair of FC layers for categorization [27]. VGG-19 and VGG-16 were also implemented and attained fidelities of 69.57% and 71.19%, respectively. For this method, 79.45% was the highest accuracy; however, this could be increased by improving the model architecture. Some of the existing techniques that are used for skin cancer detection are reported in Table 1.

## 3. Materials and Methods

In this study, we propose a robust method for skin cancer detection, i.e., *SCDet* based on a convolutional neural network (CNN). CNN is a commonly used artificial neural network (ANN), which executes the mathematical functions on feature maps recognized as convolutional [36]. CNNs work in 2 phases during the whole training process, i.e., forward-propagation and backwards-propagation. The weights assigned to the connections among consecutive layers are modified during the backpropagation phase to minimize the errors. The cost function is used to find errors by comparing the output with the ground truth [37].

There exists numerous layers in CNNs such as convolutional, pooling, batch normalization, FC, activation, and a classification layer. CNNs outperform in the problems that are related to the image data. Our proposed *SCDet* consists of 32 layers, including the image input layer as the first layer of the model, 31 hidden layers in between, along with the output layer. The proposed model’s architecture is shown in Figure 1. The dermoscopic images of skin lesions from the dermis dataset are initially provided to the image input layer of the proposed model. Convolutional layers are added to the proposed CNN for the extraction of more prominent features from the images. A batch normalization layer and ReLU layers are inserted after the pair of convolutional layers to normalize the data and prevent overfitting. After that, a pair of max pooling layers are added to summarize the features to reduce the computational cost. Finally, three fully connected layers are used to convert the data into a one-dimensional array for the detection of skin lesions. At the end, softmax and classification layers are added for the final prediction of malignant and benign tumours of skin lesions.

Max-pooling, convolution, and fully connected layers are further expanded to improve the accuracy, as shown in Figure 2. We used a pair of convolution layers after the image input layer so that our proposed model extracted the most prominent features from the images because these features are used to boost the model’s performance. Similarly, as shown in Figure 2, we used the pair of max pooling layers to reduce the computation levels and generate a smaller feature map. Finally, three fully connected layers are used for the conversion of the feature map into a one-dimensional vector.

### 3.1. Input Layer

Images are resized to a dimension of 227 × 227 pixels and passed to an image input layer. These are coloured images with a height and width of 227 each and a depth of 3, which is an RGB channel in an input layer.

### 3.2. Convolutional Layer

The convolutional layer creates feature maps that highlight the hidden features of an original image. Convolutional layers use some fixed-size filters known as convolutional filters that can be 5 × 5, 3 × 3, or even 1 × 1. Convolutional filters perform a function on input images by sliding a trained filter over the image. Weight and bias values remain the same throughout the image during the sliding of the trained filter [38]. Convolutional filters are determined during the training process and not selected manually. The convolutional filter of 2 × 2 is applied to an image of 4 × 4 pixels in Figure 3 given below. An image patch of 2 × 2 is obtained from the image, then the image patch is multiplied with the convolutional filter, and the result is obtained in the form of a matrix. The convolutional filter strides on the image with an image stride of 1 and the final matrix is the complete feature map obtained from the convolutional filter.

### 3.3. Batch Normalization Layer

The batch-normalization layer is inserted to increase the convergence of training after the convolutional layer [39]. The input of activation function is normalized with additional scaling and shifting by inserting the batch normalization layer to overcome vanishing gradient before the sigmoid/ReLU/tanh hidden layer [40]. Two channel-wise sequential operations are performed with batch normalization; the first one is normalization and the second is an affine transformation. Normalization operation includes mean and variance of batch B of data consisting of n features. Equation (1) presents the formula to calculate the mean, Equation (2) shows variance, and Equation (3) calculates the normalization.

(1)$$\mu_{B}=\frac{1}{n}\sum_{i=1}^{n}x_{i}$$(2)$$\sigma_{B}^{2}=\frac{1}{n}\sum_{i=1}^{n}(x_{i}-\mu_{B})2$$
where *B* is the mini-batch of size n for a network layer with d-dimension input;* x_i_*, $\mu_{B}$ represents the mean; and $\sigma_{B}^{2}$ represents the variance that is further used to calculate normalization. Input batch is normalized to have unit S.D. and zero mean.
(3)$$\hat{X}=\frac{X-\mu_{B}}{\sqrt{\sigma_{B}^{2}+\varepsilon }}$$

Equation (3) ε is an arbitrarily small constant for numerical stability and is added in the denominator. Then, affine transformation is applied to $\hat{X}$. Affine transformation is calculated using the equation provided below, Equation (4).
(4)$$y=ϒ.\hat{X}+\beta ,$$
where β is the shift parameter and ϒ is the learnable scale.

### 3.4. ReLU Layer

Output generated after the convolutional layer is adjusted with ReLU non-linear function to limit the output. Sigmoid and tanh can cause a problem in backpropagation, therefore we have applied ReLU as an activation function. Definition of ReLU in gradient and function is given below in Equations (5) and (6), respectively. For negative input, ReU returns 0, it returns that value for any positive value of t. Thus, the output of this function has a range from 0 to infinity.
(5)$$\frac{d}{dt}reLU\left(t\right)=\left\{\begin{array}{c} 1,t>0\\ 0,t<0 \end{array}\right.,$$


(6)
RELU (t)=max(0,t)


### 3.5. Max Pooling Layer

Feature maps generated through convolutional filter and activation function are further summarized by the max pooling process. Smaller feature maps are generated by max pooling as it reduces the size, therefore mitigating the computational load and reducing the chance of overfitting. Complexity for the next layers is reduced with downsampling, which is achieved using a pooling layer. The most important type of pooling is max pooling which returns only a rectangular sub-region of an image with a maximum value of the sub-region. The largest or maximum value of each patch of every feature map is calculated with the max pooling operation. The results are down-sampled and highlight the most present features in the patch, the process of max pooling is shown in Figure 4.

### 3.6. Fully Connected Layer

Data are transformed into a one-dimensional vector after pooling and convolutional layer and provided to the FC layers. Connection weights are multiplied by previous layer data and with an added bias value. The operation performed by a fully connected layer is represented in Equation (7).
(7)$${\mathrm{Fc}}_{1}=f\;(b+\sum_{r=1}^{n}w_{1},r\;*\;o_{r}),$$
where *w* shows the weight vector; *o* represents the input vector of the r^th^ neuron; b represents bias value; and f is an activation function.

### 3.7. Softmax Layer

For multiclass classification softmax layer is used. Selection of activation function performed by softmax is provided in Equation (8) below.
(8)$$s\left(v_{i}\right)=\frac{e^{v}i}{\sum_{j=1}^{n}e^{vj}},$$
where *S* is a softmax function of input vector *v*, *n* shows no. of classes; e^v^i represents the standard exponential function for the input vector; and e^vj^ is the standard exponential function for output vector. The detail of layers used in architecture is given in Table 2.

Process of skin lesion detection into benign and malignant classes is presented in Figure 5. Initially, images are provided to the image input layer, then features are extracted using convolutional layer. After that, extracted features are normalized and provided as input to the ReLU layer. ReLU layer regularizes the features, which helps the model to minimize the problem of overfitting. After that, features are summarized using max pooling and finally classified as benign and malignant tumours by classification layer.

## 4. Experimentation Methods

### 4.1. Dataset Used

Dermis dataset images are collected from the Kaggle website, which is publicly available [41]. Benign and malignant are the two classes that are used from the dermis dataset. In total, 75% of the images of both classes have been used to train the model, and the remaining 25% of images have been used for validation purposes. The images comprised 1000 in total and had two classes, i.e., benign and malignant. Moreover, 500 images are benign and 500 belong to the malignant class, having dimensions of 600 × 450 pixels. Pre-processing is performed on images to resize them to 227 × 227 pixels. Samples of the images of benign and malignant classes are provided in Figure 6.

### 4.2. Environmental Setup

First, 750 images out of 1000 are selected randomly for the training process and 250 are selected for validation of the model. We choose SGD with momentum as an optimizer with an initial-learning rate of 0.01, learn-rate drop factor of 0.2, and learn-rate drop period of 5. Mini-batch size is adjusted to 64 and Max-epoch is 20. The training process of the model with the loss and accuracy of the model observed in MATLAB is presented in Figure 7.

Recall, precision, sensitivity, accuracy, and specificity are the performance matrices of the proposed CNN model. The percentage of benign images that the system recalled is the recall of the model. The recall is computed as the fraction of malignant lesions of skin that are correctly classified to all positive images of malignant class and false-classified benign class images, see Equation (9)
(9)Recall=TP/(FN+TP),

Specificity is to correctly identify benign class images, also called a true negative rate and calculated as the proportion of generally negative samples to given negative results by model. Equation (10) is used to calculate specificity.
(10)Specificity=TN/(FP+TN),

Percentage of accurately classified images is the precision that is provided by the proposed CNN model. The number of images that are generally positive is divided by the total number of predicted positive classes and calculated using Equation (11) shown below.
(11)Precision=TP/(FP+TP),

Sensitivity is the ability to correctly identify a malignant class by the model. Sensitivity, sometimes referred to as true positive rate TPR, is the calculated proportion of positive samples that give positive results.
(12)Sensitivity=TP/(FN+TP),

All correct predictions provided by the proposed CNN model reflect the accuracy. The number of predictions that are correct divided by the total number of predictions provides the accuracy.
(13)accuracy=TN+TP/(FP+TN+TP+FN),

Evaluation metrics are presented in Table 3.

The percentage of all correctly classified images represents the precision of the model [42]. It is observed from the confusion matrix that the proposed CNN provides high accuracy as the diagonal of the confusion matrix has high values in Figure 8. The result of the confusion matrix is also represented in terms of percentage accuracy of both classes, benign and malignant, used in our dataset. In the Figure 8 confusion matrix, there are 125 images of the benign class that our proposed model correctly classified and there was only one case that the proposed model classified as in the benign class but actually the image belonged to the malignant class. Similarly, 124 images were correctly classified by the model as the malignant class, and there was no such case where the proposed model classified it as malignant and the image belong to the benign class. So, it is clearly observed that out of 250 images, 249 images are correctly classified by the model and only one image was not classified correctly by the proposed model.

### 4.3. Validation

The proposed model was tested on the images of another dataset that were not used during training, as well as the validation process of our model. Proposed CNN performs best in the classification of these images as benign or malignant. Proposed CNN provides an accuracy of 85% on the HAM1000 dataset. The HAM1000 dataset is used for validation purposes, which is collected from the Kaggle website [43]. We use two classes for classification, melanoma and not melanoma as malignant and benign. Training accuracy and loss of proposed CNN on the HAM1000 dataset are presented in Figure 9. From Figure 8, it is clearly observed that proposed CNN provides 85% accuracy on the HAM1000 dataset, which is used for the validation of the proposed CNN. Although this accuracy is not good enough, it is much better than the accuracy of the pre-trained model i.e. AlexNet and SqueezeNet as shown in Figure 10 and Figure 11, respectively.

The proposed model is also validated by training on another dataset and it is also compared with AlexNet, SqueezeNet, and VGG16.

### 4.4. Comparison with Existing Pre-Trained Models

#### 4.4.1. Alex Net

In Ref. [44], the research authors proposed a model AlexNet that was trained on 1000 different classes. S. Sadhana and R. Mallika in [45] used AlexNet for the detection of diabetic retinopathy. We utilized AlexNet for skin lesion classification as benign and malignant for comparison but pre-trained AlexNet provides an accuracy of only 90%. However, 99.6% accuracy is obtained by our proposed CNN, which is much better than pre-trained AlexNet. Training accuracy of AlexNet is presented in Figure 10.

#### 4.4.2. SqueezeNet

In Ref. [46], authors proposed a model SqueezeNet that provides similar accuracy to the AlexNet on the ImageNet with 50x fewer parameters. K. Nakamichi and H. Lu focused on attaining significant results on circulating tumour cells (CTC) classification using fluorescence microscopy images with pre-trained CNN model SqueezeNet [47]. Similarly, K. N. Akpinar, S. Genc, and S. Karago identified the presence of disease in the chest from chest X-ray using SqueezeNet [48]. The complexity of network is high and classification accuracy is low as compared to our proposed model. SqueezeNet has 68 layers while our model has only 32 layers, so the complexity of SqueezeNet is high and it is slow compared to our model. A 99.6% accuracy is achieved by our proposed model, whereas SqueezeNet provides only 83% accuracy, which is very low. Figure 11 shows training accuracy of the deep learning pre-trained model SqueezeNet.

#### 4.4.3. VGG16

In Ref. [49], the authors proposed a model that improves the accuracy of classification-related problems by increasing the depth of the model. Panthakkan, S. M. Aznar used X-rays of lungs and VGG16 for the prediction of COVID-19 into binary classes, i.e., positive COVID-19 and negative COVID-19 [50]. H. Aung also used VGG16 with the combination of the YOLO algorithm to detect the face from real-time live video [51]. However, VGG16 faced an issue of vanishing gradient. The accuracy of VGG16 for the classification of the skin lesion is low compared to our *SCDet*, as our model provides 99.6% accuracy whereas VGG16 provides about 80%. A comparison of various models with *SCDet* is listed in Table 4.

### 4.5. Comparison with Machine Learning Techniques

In this experiment, we compare the performance of *SCDet* with ML-based methods. The purpose of the [52] study was melanoma detection from skin lesions using a machine learning model. Homogeneity, correlation, energy, and contrast features were used. PH2 dataset was used and 3-fold cross-validation was applied to divide the data into training and validation. An SVM classifier was utilized for the prediction of skin lesions that provided 96% accuracy. MS.H.R. Mhaske classified melanoma class of skin cancer with one of the supervised learning techniques, i.e., K-means and two supervised learning techniques SVM and Neural networks. A low-pass and a high-pass filter were applied and then segmentation was performed to achieve better results [53]. The unsupervised technique, K-means algorithm, obtained 52.63% accuracy, whereas supervised learning technique, i.e., neural network gained 75%. The support vector machine performed best among these, providing 80–90% accuracy. Authors in [15] identified malignant and benign classes using an ML model, named support vector machine. Image preprocessing was performed to improve the quality of the image and to eliminate hair, noise, and skin colour. GLCM and HOG feature extractor was applied to extract the features from images. A 97.8% accuracy was obtained using an SVM. A comparison of accuracy for *SCDet* with ML techniques is presented in Figure 12.

### 4.6. Comparison with Segmentation-Based Techniques

In this experiment, we compared the performance of *SCDet* with the segmentation-based methods. The aim of [54] was to recognize skin cancer using segmentation and then feature extraction. First, authors utilized image pre-processing using median filter to reduce the noise from images. The size they used for median filter was 5 × 5. Further, morphological operations such as dilation and translation were used to remove the skin hair and colour from images. The authors used threshold technique to obtain the region of interest from the image. They attained classification accuracy of 90.83%. Manu Gofal [18] developed a technique using boundary segmentation employing dermoscopic images through DL method. ISIC 2017 and PH2 datasets were utilized for lesion segmentation. Image preprocessing was utilized to enhance the visual features attained from the images. The proposed method achieved 98% sensitivity and 93% accuracy. The aim of [55] was to automate the process of classification for skin cancer by employing deep learning. The U-Net was used for significant skin cancer localization. In the segmented part, the images contained the disconnected objects; therefore, to enhance the region of interest, authors used post-morphological operations. The noise and extra hairs were removed from the images. Region-mapping functions were utilized for selection of region with the maximum area. The proposed technique attained 96% accuracy and 98% specificity. The comparative analysis in form of a plot is shown in Figure 13.

### 4.7. Comparison with Existing DL-Based Skin Cancer Detectors

N. Rezaoana categorized 9 classes: MEL, AKIEC, NV, VASC, BCC, BKL, DF, seborrheic keratosis, and squamous carcinoma with a CNN model. Rotation, flip, shear, etc. were the data augmentation techniques utilized to enhance the amount of data for the training purpose. Moreover, Conv. layers were utilized to extract features and max-pooling was used to minimize the dimensionality [27]. Various metrics such as F1-score, precision, accuracy, and recall were utilized to assess the performance. They compared the results with VGG-19 attaining an accuracy of 69.57% and for VGG-16 at 71.19%. Whereas our proposed method attained 79.45% accuracy. The aim of [56] was the identification of types of skin lesions by employing an artificial neural network. PH2 dataset was used consisting of only 40 images of melanoma and 160 benign. Pre-processing was performed on images that were used to minimize the noise and segment the area. The shape and centroid features were utilized achieving 98% accuracy. Y. Filali, developed the classification technique by decomposing the images of skin lesions into texture and object components. To attain the region of interest, segmentation was employed on objects and then textures and segmented areas were combined. Feature extraction was performed using a Convolution layer. A pooling layer just followed by the convolution layer was employed to minimize the spatial size and then fully connected layers were applied. Softmax activation function followed by final classification layer was used as an activation function to classify Melanoma as cancerous and Nevus as a non-cancerous class. A 93.50% accuracy was attained, which could be improved [26]. Varma, in [57], introduced a method named SLDCNet for skin cancer detection using image pre-processing and full-resolution CNN. The proposed model provides 99.92% accuracy, however, the SLDCNet model uses a hybrid approach with pre-processing, which makes it computationally expensive compared to our SCDet model. Additionally, SCDet is more user friendly and easier to use, making it a simple alternative to the more complex SLDCNet model. A comparative analysis with existing methods is given in Figure 14, and the results are reported in Table 5.

## 5. Conclusions

We propose an effective architecture, named *SCDet*, for skin lesions classified as benign and malignant in this study. To train our model, the Dermis dataset was utilized from the Kaggle, which is publicly available [41]. There exists in total 1000 images comprising 500 images belonging to the benign class and 500 to the malignant class. A pair of convolutional layers were used to extract hidden patterns from the tumourous samples. Batch normalization was added for the normalization of features. A ReLU layer was employed in the proposed network that is faster than the sigmoid activation function. Then, a pair of the max-pooling layers along with three FCs and softmax is used before the classification layer. The proposed methodology attained 99.6% accuracy, 100% recall, 100% sensitivity, 99.2% specificity, and 99.2% precision. The proposed model is also evaluated on another dataset and provides an accuracy of 85% on that dataset. Moreover, we also compared the results of *SCDet* with VGG16, AlexNet, and SqueezeNet, and observed that the proposed methodology provides higher accuracy among these pertained networks effectively. In the future, we aim to fine-tune our network for subclasses of benign and malignant skin cancer, e.g., Markel cell carcinoma, squamous cell carcinoma, Kaposi sarcoma, Basal cell carcinoma, etc.

## Figures and Tables

**Figure 1 diagnostics-13-01824-f001:**
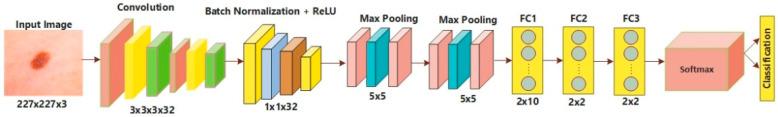
The Proposed CBIR.

**Figure 2 diagnostics-13-01824-f002:**
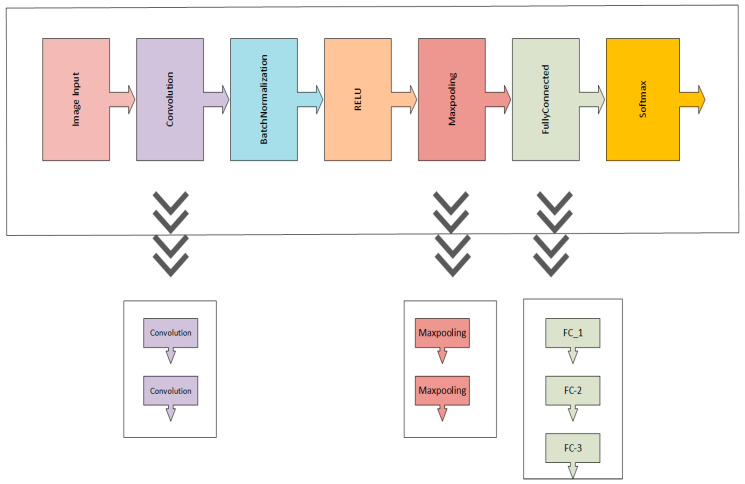
Expanded Layers of proposed model.

**Figure 3 diagnostics-13-01824-f003:**
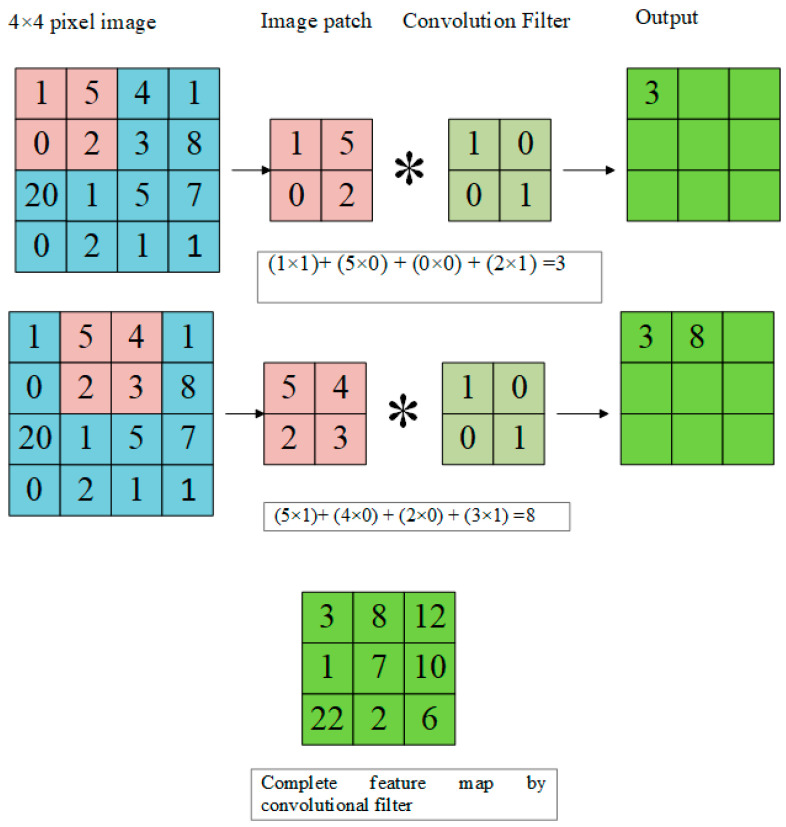
Convolution Filter feature map on image.

**Figure 4 diagnostics-13-01824-f004:**
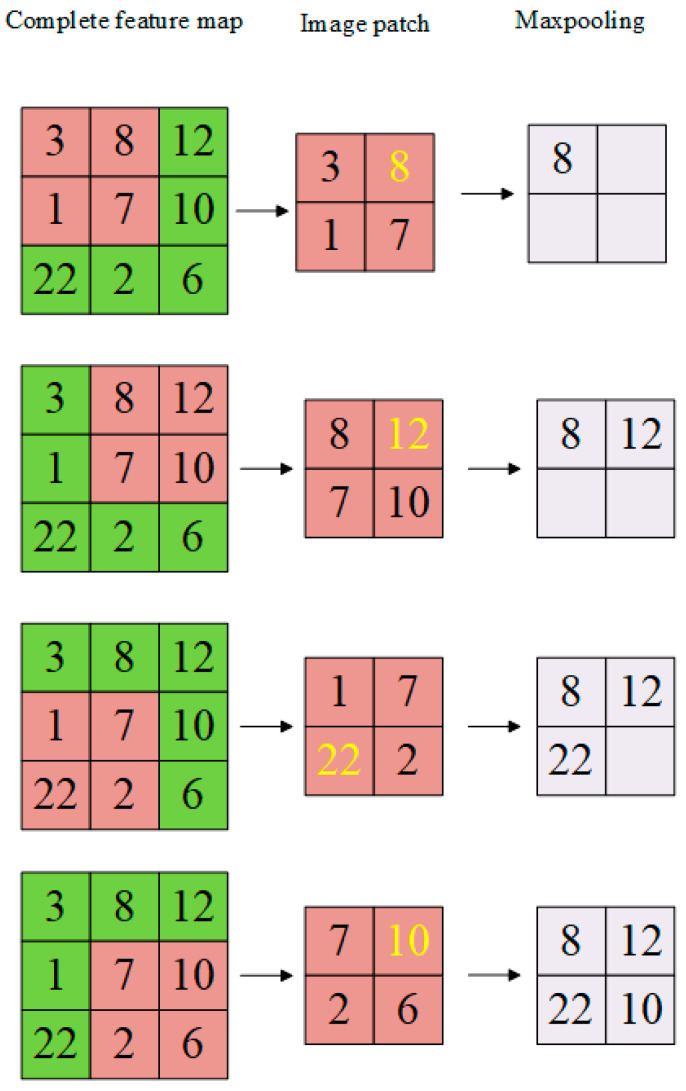
Max Pooling Process.

**Figure 5 diagnostics-13-01824-f005:**
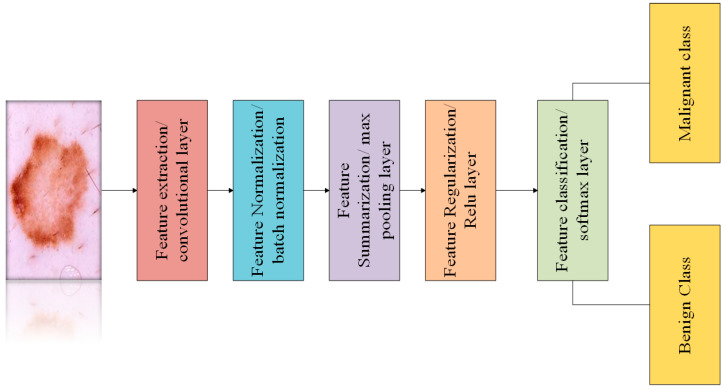
Skin Lesion Detection Process.

**Figure 6 diagnostics-13-01824-f006:**
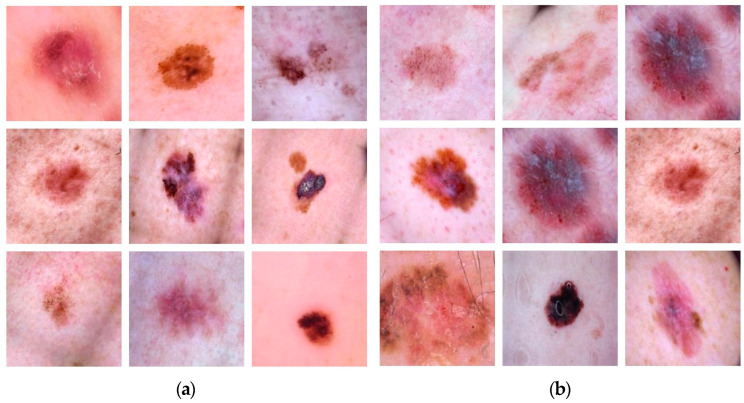
Samples of Images used in Dataset. (**a**) Benign, (**b**) Malignant.

**Figure 7 diagnostics-13-01824-f007:**
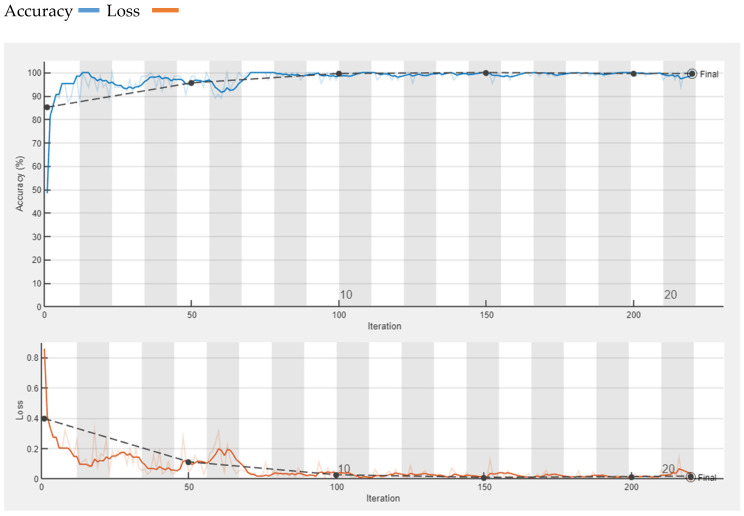
Accuracy and Loss of proposed CNN.

**Figure 8 diagnostics-13-01824-f008:**
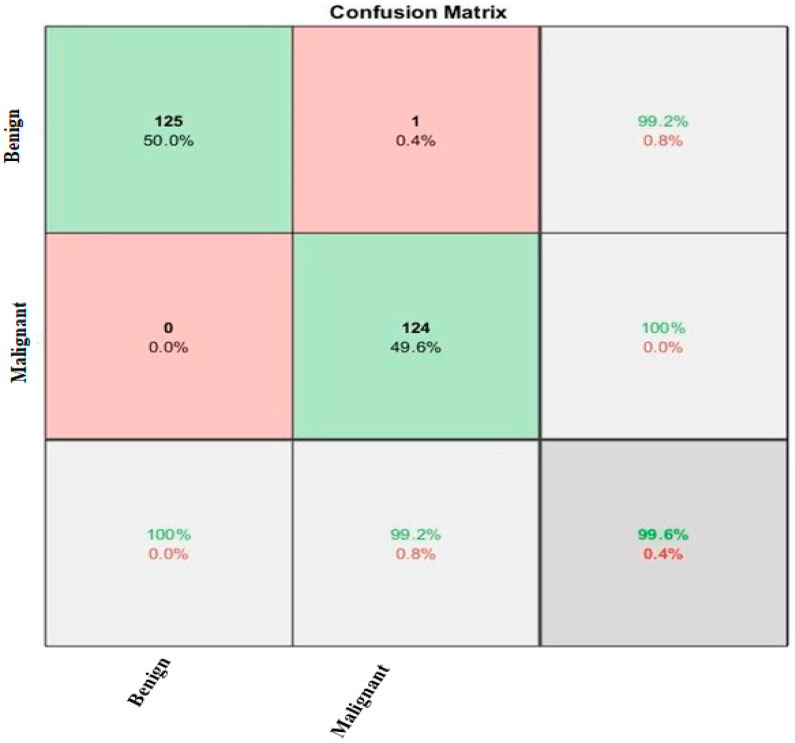
Confusion Matrix.

**Figure 9 diagnostics-13-01824-f009:**
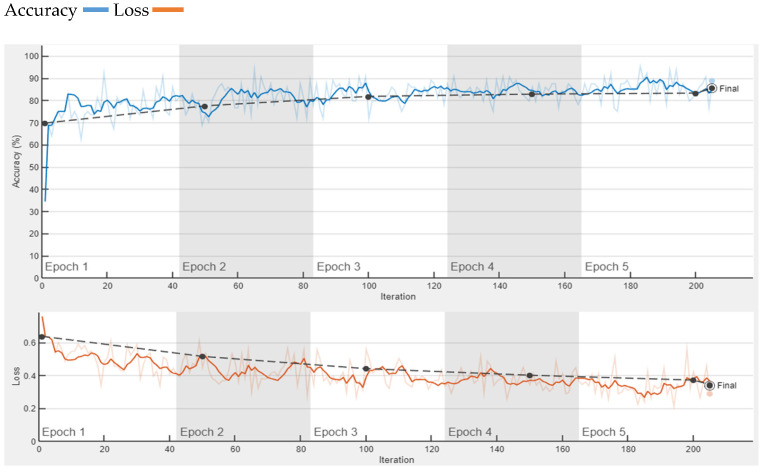
Accuracy of proposed CNN on HAM1000 Dataset.

**Figure 10 diagnostics-13-01824-f010:**
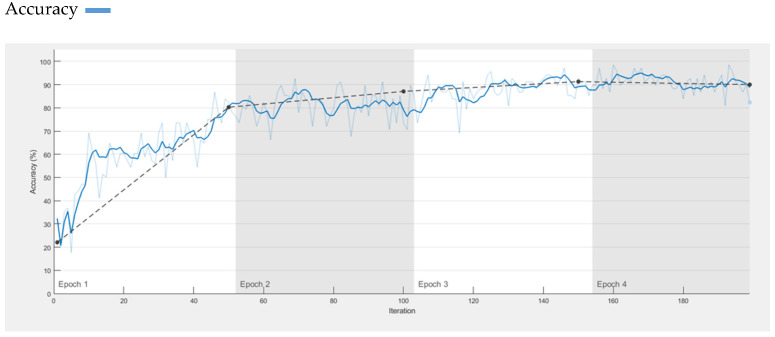
Accuracy of AlexNet on Dermis Dataset.

**Figure 11 diagnostics-13-01824-f011:**
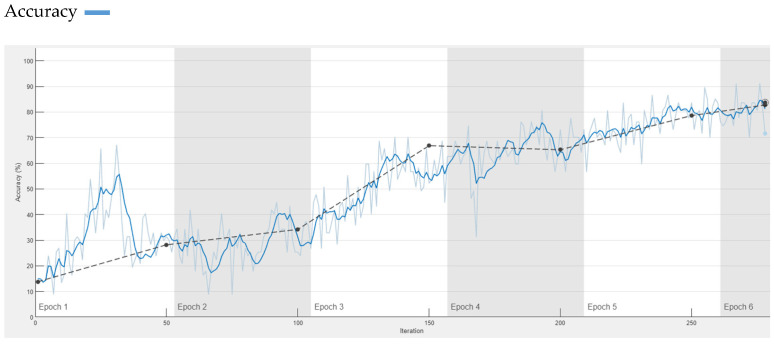
Accuracy of SqueezeNet on Dermis Dataset.

**Figure 12 diagnostics-13-01824-f012:**
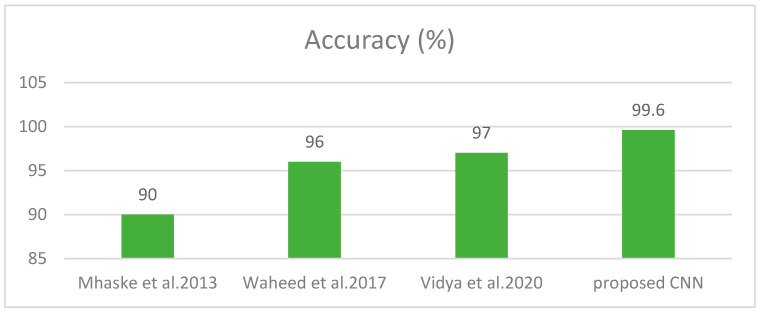
Comparison with Machine Learning Techniques [15,52,53].

**Figure 13 diagnostics-13-01824-f013:**
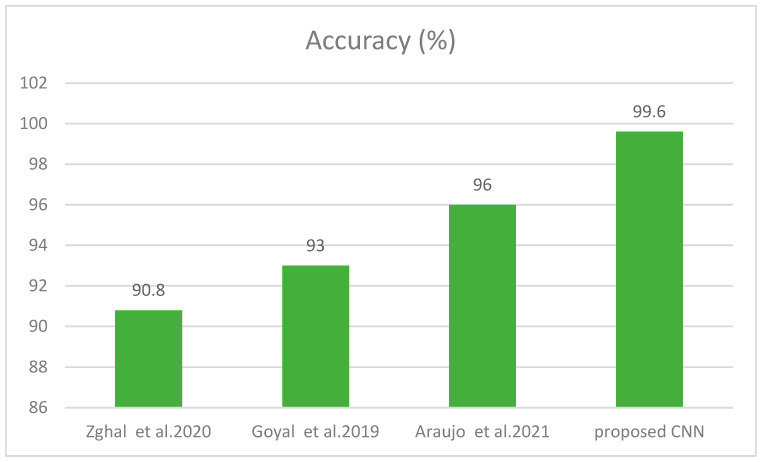
Comparison with Segmentation Techniques [18,54,55].

**Figure 14 diagnostics-13-01824-f014:**
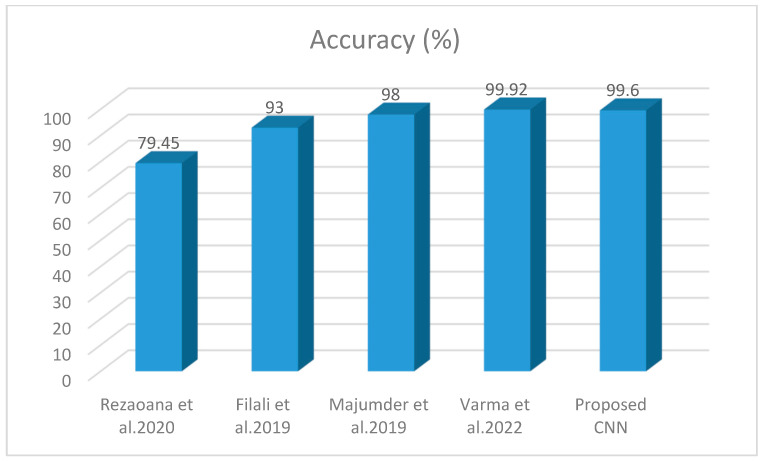
Comparison with Deep Learning Techniques [26,27,56,57].

**Table 1 diagnostics-13-01824-t001:** Comparison of Related Work.

Ref	Year	Classes of Skin Lesions	Model Type	Model	Activation Function	Dataset Used	Accuracy
[28]	2019	Melanoma, Non-Melanoma	supervised	CNN	ReLU	ISIC 2017, PH2	95%
[29]	2020	NV, VASC, DF, BCC, MEL, AKIEC, BKL	supervised	CNN	ReLU	ISIC 2019	96%
[30]	2020	Melanoma, Nevus, Seborrheic Keratosis	supervised	CNN	softmax	ISIC2018, HAM10000	86%
[20]	2020	Binary	supervised	CNN	ReLU	ISIC	80%
[31]	2020	Melanoma, common Nevus, atypical Nevus,	supervised	CNN	softmax	PH2	95.0%
[32]	2020	NV, DF, BKL VASC, MEL, BCC, AKIEC	supervised	CNN	ReLU	HAM1000	90%
[33]	2021	Benign and malignant	supervised	CNN	SIGMOID	HAM10000	90.93%
[34]	2021	Binary	supervised	CNN	-	Dermis	95%
[35]	2022	NV, DF, MEL, AKIEC, VASC, BCC, BKL	supervised	CNN	-	PH2	95%

**Table 2 diagnostics-13-01824-t002:** Layer architecture of proposed model SCDet.

Type	Channel/Stride	Learnable	Activation
input	-	-	227 × 227 × 3
4 × [Conv_1]	Stride [1 1]	Weights 3 × 3 × 3 × 32	227 × 227 × 32
	Padding same	Bias 1 × 1 × 32	
4 × [Conv_2]	Stride [1 1]	Weights 3 × 3 × 32 × 32	227 × 227 × 32
	Padding same		
		Bias 1 × 1 × 32	
4 × [Batchnorm_1]	32 channels	Scale 1 × 1 × 32	227 × 227 × 32
		Offset 1 × 1 × 32	
2 × [FC-1]			
4 × Relu_1	-	-	227 × 227 × 32
4 × [Maxpool_1]	Stride [1 1]	-	227 × 227 × 32
	Padding same		
	5 × 5 max pooling		
4 × [Maxpool_2]	Stride [1 1]	-	227 × 227 × 32
	Padding same		
	5 × 5 max pooling		
Fc_2	-	Weights 2 × 10	1 × 1 × 2
		Bias 2 × 1	
Fc_3	-	Weights 2 × 2	1 × 1 × 2
		Bias 2 × 1	
Fc_4	-	Weights 2 × 2	1 × 1 × 2
		Bias 2 × 1	
softmax	-	-	1 × 1 × 2
Class output	-	-	1 × 1 × 2

**Table 3 diagnostics-13-01824-t003:** Evaluation Metrics of the proposed model SCDet.

Method	TP	FP	FN	TN	Recall	Specificity	Precision	Sensitivity	Accuracy
Proposed CNN	49.6%	0%	0.4%	50.0%	100%	99.2%	99.2%	100%	99.6%

**Table 4 diagnostics-13-01824-t004:** Comparison with pre-trained models.

Method	AlexNet	SqueezeNet	VGG16	Proposed CNN on HAM1000	Proposed CNN
**Accuracy**	90%	83%	80%	85%	**99%**

**Table 5 diagnostics-13-01824-t005:** Comparison with ML, DL, and Segmentation Techniques.

Ref.	Model Type	Precision	Specificity	Recall	Accuracy
[52]	ML	-	84%	97%	96 %
[15]	ML	-	85%	86%	97%
[55]	Segmentation		98%	93%	96%
[27]	DL	76%	-	78%	79%
[54]	Segmentation	-	97%	87.5%	90%
[56]	DL	-	98.75%	95%	98%%
[53]	ML	-	-	-	90%
[26]	DL	-	-	-	93.5%
[18]	Segmentation	-	97%	89%	94%
**Proposed** **CNN**	DL	**99.2%**	**99.2%**	**100%**	**99.6%**

## Data Availability

As the authors have utilized openly accessible datasets, which are elaborated in the “experimental results and discussions” segment of this article, data sharing is not relevant to this article. If you require additional information, kindly reach out to the authors.

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
