# Peer review of "SCDet: A Robust Approach for the Detection of Skin Lesions"

_diagnostics, 2023, doi:10.3390/diagnostics13111824_

Round 1

Reviewer 1 Report

The paper entitled, 'SCDet: A Robust Approach for the Detection of Skin Lesion' by Shahbaz Sikandar et al. proposed a novel and robust method for detecting skin lesions using a 32-layer convolutional neural network (CNN) called SCDet. The results of their experiments demonstrate the effectiveness of their method in detecting skin cancer with high accuracy and robustness.

The paper includes a comprehensive evaluation of the proposed algorithm, with experiments on various datasets and comparisons with existing methods. Moreover, the paper's structure is well-organized, making it easy to follow along with the authors' arguments and methodology. The results presented in the paper are impressive and demonstrate the effectiveness of the proposed algorithm in comparison to other state-of-the-art techniques. The authors did an excellent job of showcasing the most relevant results and conclusions, emphasizing the importance and novelty of their work. The paper's structure is also well-organized, with a logical flow of information and easy-to-follow figures and tables. The language is clear and concise, making it easy for readers to understand the methodology and results.

Overall, this article makes a significant contribution to the field of medical image analysis and has the potential to have a positive impact on the diagnosis of skin cancer and it may be accepted for publication in its current form.

Reviewer 2 Report

In this manuscript, the authors reported a deep learning based model SCDet for the classification of  benign and malignant skin cancer. The performance of the model is impressive, with precision 99.2%, recall 100%, Sensitivity 100%, Specificity 99.20%, and accuracy 99.6%. Overall this work was presented very well though each layer seems over detailed. The only concern I have is about the results listed in Table IV. It seems that the configuration for Alex net Squeeze net had some problems, because the results showed that the performance of  two models is no difference from random guess(50% accuracy). I would suggest that the authors can go back to check the configuration.  

Author Response

  1. In this manuscript, the authors reported a deep learning based model SCDetfor the classification of  benign and malignant skin cancer. The performance of the model is impressive, with precision 99.2%, recall 100%, Sensitivity 100%, Specificity 99.20%, and accuracy 99.6%. Overall this work was presented very well though each layer seems over detailed. The only concern I have is about the results listed in Table IV. It seems that the configuration for Alex net Squeeze net had some problems, because the results showed that the performance of  two models is no difference from random guess(50% accuracy). I would suggest that the authors can go back to check the configuration.  

Response:Thank you for your feedback. We have made further adjustments to the network parameters based on your suggestions, and Alexnet provides 90% accuracy while Squeezenet provides 83%. Actually initially we trained Alexnet and squeezent with their default training parameters that’s why they were providing only 50% accuracy.

Reviewer 3 Report

The authors propose the theme of Detection of Skin Lesion using Deep Learning, which has been addressed in the specialized literature. The novelty consists in the approach using Deep Learning and in the accuracy obtained of almost 100% compared to similar results reported in specialized literature. I congratulate the authors for the article and I saw that some of the authors published another article on the same topic (Skin Lesion Detection Using Hand-Crafted and DL-Based

Features Fusion and LSTM) in Diagnostics magazine.

From my point of view, I want from the authors a comparison between the proposed method and the article "SLDCNet: Skin lesion detection and classification using full resolution convolutional network-based deep learning CNN with transfer learning". In this article, the authors report similar results with almost 100% recognition (Cite article - Results - The extensive simulation results shows that proposed SLDCNet resulted in a classification accuracy of 99.92%, sensitivity of 99%, and specificity of 99.36%, respectively).

I ask the authors to clarify the differences between the method proposed by them and the method from the previously mentioned article.

Author Response

  1. The authors propose the theme of Detection of Skin Lesion using Deep Learning, which has been addressed in the specialized literature. The novelty consists in the approach using Deep Learning and in the accuracy obtained of almost 100% compared to similar results reported in specialized literature. I congratulate the authors for the article and I saw that some of the authors published another article on the same topic (Skin Lesion Detection Using Hand-Crafted and DL-Based.

From my point of view, I want from the authors a comparison between the proposed method and the article "SLDCNet: Skin lesion detection and classification using full resolution convolutional network-based deep learning CNN with transfer learning". In this article, the authors report similar results with almost 100% recognition (Cite article - Results - The extensive simulation results shows that proposed SLDCNet resulted in a classification accuracy of 99.92%, sensitivity of 99%, and specificity of 99.36%, respectively).

I ask the authors to clarify the differences between the method proposed by them and the method from the previously mentioned article.

Response:Thanks for your comment. AS per your suggestion we added the comparison of our model with the Suggested model in the comparison with existing techniqes of DL method section ans in plot as well.

“Varma et al. in [57], introduces a method named SLDCNet for skin cancer detection by using image pre-processing and full resolution CNN. Proposed Model provides 99.92% accuracy however, The SLDCNet model uses a hybrid approach with pre-processing, which makes it computationally expensive compared to our SCDet model. Additionally, SCDet is more user-friendly and straightforward to use, making it a simpler alternative than the SLDCNet model.”

Reviewer 4 Report

The authors have validated a new convolutional neural network (CNN)-based system, named SCDet, as a robust approach to the diagnosis of skin lesions, especially skin tumors.

The results are very impressive and seem to be superior to previous reports.

Regarding Figure 6, there are only 16 images in a row, but I would like to see a legend for each diagnosis name, if possible. And were the results benigh or malignant? This might be useful to non-dermatology readers, but what do you think?

Author Response

The authors have validated a new convolutional neural network (CNN)-based system, named SCDet, as a robust approach to the diagnosis of skin lesions, especially skin tumors.The results are very impressive and seem to be superior to previous reports.

Regarding Figure 6, there are only 16 images in a row, but I would like to see a legend for each diagnosis name, if possible. And were the results benigh or malignant? This might be useful to non-dermatology readers, but what do you think?

Response:Thanks for your comment. As per yor suggestion we updated the Figure 6 as suggested.

Round 2

Reviewer 2 Report

The authors have addressed my questions and I have no more comments on that.